# Lending Technologies, Firm Characteristics and Small Business Efficiency in South Africa

**Edson Mbedzi \*** and **Munacinga Simatele**

Department of Economics, University of Fort Hare, Private Bag X9083, 50 Church Street, East London 5200, South Africa
* Correspondence: edsonmbedzi@gmail.com

**Abstract:** Internal factors of Small, Micro and Medium Enterprises (SMMEs) determine their technical efficiency, while external funding characteristics improve the quality of internal factors. Since the type of lending institutions and lending technologies primarily influence the lending decisions of financial institutions, firms' technical efficiency may be linked to such external factors. Literature on determinants of the technical efficiency of SMMEs mainly focuses on internal factors excluding the financial access paradigm which stifles the effectiveness of internal factors on technical efficiency. Based on a sample of 321 randomly selected SMMEs from Eastern Cape Province in South Africa, the study measures technical efficiency using Data Enveloping Analysis and differentiates technical efficiency among firms using Post Hoc Test Pairwise Comparisons derived from factorial ANOVA. Both main and interaction effects were captured in the analysis. Our results, which pinpoint four main findings, show technical efficiency paths followed by firms vary significantly as a result of both internal and external factors. In particular, the effects of other factors are amplified by race. As a consequence, three main contributions emerge from the study.

**Keywords:** technical efficiency; SMME structure; lending institution structure; DEA; factorial ANOVA

## 1. Introduction

Small, Micro and Medium Enterprises (SMMEs) are engines of growth in most economies. The enterprises account for almost 90% of businesses in both leading and developing economies through job creation, employment, tax provision and contribution to gross domestic product. Nevertheless, low technical efficiency impairs their progress. However, the evidence shows that the level of attrition among SMMEs in Africa is quite high (UN 2021). Technical efficiency is a predictor of business failure (Kumar 2022). World Bank's SMMEs surveys in South Africa showed SMMEs have low technical efficiency due to high factor unit costs compared to those in peer countries like Brazil, Chile and Argentina (World Bank 2010). As a result, their inability to operate on or close to the production frontier enlarges the productivity gap between SMMEs and other firms within the economy. The implication is that growth in technical efficiency is necessary to foster the growth of SMMEs. Several studies have investigated the determinants of firm technical efficiency. These studies have identified various factors that affect technical efficiency within SMMEs.

One of the most prominent factors identified in the literature is firm size. Authors like Jovanovic (1995) and Agostino and Trivieri (2019) have shown theoretically that due to firm learning, larger firms are likely to be more efficient. The empirical literature shows mixed results. Several firms confirm this relationship. For example, the results of IKram et al. (2016) and Fahmy-Abdullah et al. (2021) show a positive relationship between firm size and technical efficiency. However, some results show that ignoring the heterogeneity among SMMEs can result in the wrong conclusion that all larger firms are more efficient than smaller ones. Some results show that firm size can have a negative relationship with efficiency (Le and Harvie 2016). Others, like Batra and Tan (2003) and Charoenrat and

Harvie (2014), show that the effect of size is conditional on various factors including the sector and investment in human capital. Batra and Tan (2003) find that some small firms are more efficient than larger ones once conditioned on the sector of operation and investment in human capital. Similarly, Charoenrat and Harvie (2014) find variation in the relationship and show that although small firms were generally less efficient than medium firms over ten years, small firms in various sectors were more efficient than medium-sized firms. They show that the efficiency varied depending on the industry characteristics and financing.

Ownership structure affects the efficiency of the firm. The first channel of transmission is risk-taking. The ownership structure is one of the channels of influence that has also been flagged as an important factor in determining technical efficiency in SMMEs. Ownership type has been measured in various ways including a distinction between private and public (Margono and Sharma 2006), foreign and domestic (Goldar et al. 2003) as well as by type of registration and enterprise (Padmavathi 2019). The results show that foreign-owned films are more technically efficient than domestic-owned firms (Charoenrat and Harvie 2014). In addition, the technical efficiency of state-owned enterprises was less technically efficient than private-owned enterprises (Charoenrat and Harvie 2013). Related, (Padmavathi 2019) showed that sole-owned enterprises tended to be less efficient than other types of enterprises.

Our paper introduces three additional variables; namely, types of lenders, types of lending technologies and race. The first two variables are based on the source of funds. Contemporary literature mainly focuses on the effect of credit on technical efficiency. This literature suggests that credit can enhance efficiency when tailored to the clients' needs. In the agricultural sector where most of this literature is, it is argued that access to credit allows farmers to be more willing to adopt newer technologies that improve efficiency. A similar argument can be presented for other small and medium enterprises. Simatele and Mbedzi (2021) show that smaller firms are more likely to be price rather than quantity rationed relative to larger firms. This rationing can be attributed to the high information opacity of SMEs. As a result, the type of lender and lending technology is key in sourcing capital and performance. Lending technologies that rely on soft information such as relationship lending are likely to offer more favourable interest rates, resulting in better firm performance. For this reason, the source of funding rather than the amount received by SMMEs is likely to have a great impact on technical efficiency. Some literature in this area focused on the impact of bank funding on technical efficiency in SMMEs (UNCTAD 2001). We argue that the source of funding is a key factor in improving the technical efficiency of SMMEs. The literature shows that certain types of lenders such as banks may characteristically ration SMMEs from credit markets and offer lending at higher rates which may not be appropriate to the needs of the SMMEs (Simatele and Dlamini 2020).

Related to this, there is evidence that levels of credit rationing differ by the type of lending technology used (Mbedzi and Simatele 2020). A lending technology is defined in the paper by Berger and Udell (2006) as a combination of information sources loan structure and associated screening and monitoring mechanisms used by lenders. Moreover, lending technologies are influenced by information asymmetries which through the signalling channel can affect loan pricing and hence performance (Udell 2009; Motta and Sharma 2020). For example, Baas and Schrooten (2006) argue that interest rates tend to vary by lending technique. In addition, lending technologies vary by bank ownership and lender size (Badulescu and Badulescu 2010; Berger and Udell 2006) both of which have been shown to affect efficiency. As a result, the type of lending technique is expected to affect technical efficiency (Paxton 2007; Agostino et al. 2018). Several studies on technical efficiency have been conducted in South Africa. However, these have mainly focused on sector-by-sector analysis, such as technical efficiency in the banking sector, agricultural cooperatives and local government municipalities, with little focus on SMMEs (Akinloye et al. 2010; Gwebu and Matthews 2018; Mazorodze 2019; Mbonigaba and Oumar 2016; Xaba et al. 2018) while the few targeting SMMEs largely assess the impact of internal factors of firms on technical efficiency only (Castillo et al. 2012; Gwebu and Matthews 2018; Mthimkhulu

and Aziakpono 2016). Some studies (Tenaye 2020) show that technical efficiency can be amplified by external factors, for example, the technical efficiency of small-scale farmers in Ethiopia was higher for farms issued with land certification than those without based on the national land fragmentation policy. SMMEs in South Africa just like in any other developing economy, it is well documented the greatest challenge they faced is access to finance (Ayyagari et al. 2011; Beck and Cull 2014; Beck et al. 2009; Makina et al. 2015; Uchida et al. 2012). Therefore, it is logical to suggest that a more holistic approach to determining drivers of the technical efficiency of firms should consider both internal and external factors of the firm, particularly external factors related to access to finance (Rahaman 2011).

The level of information available on a firm typically depends on these various internal and external factors as assessed by the lender. For example, the availability of information on the firm's performance will determine which technology is used by the lender to screen for loan eligibility. In addition, other factors such as the value and type of assets and number of employees can influence lending technologies and hence efficiency (Amornkitvikai et al. 2014; Charoenrat and Harvie 2017; Castillo et al. 2012; Gwebu and Matthews 2018). Nevertheless, the role played by lending technologies in influencing firms' technical efficiency has received little attention in the literature. Most studies investigate the level of efficiency in the economy (Barchue and Aikaeli 2018; Ismail et al. 2014; Le and Harvie 2016; Mohamad et al. 2010), identify internal factors influencing firms' technical efficiency (Charoenrat and Harvie 2013, 2014, 2017; Amornkitvikai et al. 2014; Berger and Udell 1995) and how different measures of technical efficiency affect the results (Moyo 2018; Barchue and Aikaeli 2018). Thirdly, we added race as the final variable. In the South African context, the race of an individual is an important factor that influences economic activities among different production units including the flow of funds from financial institutions to individuals and businesses, hence influencing business activity along racial lines (Gwebu and Matthews 2018; Mthimkhulu and Aziakpono 2016). As a result, it is expected that race affects the technical efficiency of firms. To that point, national policies such as the Broad-Based Black Economic Empowerment (B-BBEE) policy were enacted to deal with the effects of racial discrimination on economic activity across the whole economy.

This paper, therefore, adds the role of lending technologies as a key factor in understanding small firm efficiency. Furthermore, we check how lending technologies interact with types of lenders and other key factors such as firm type and size. In addition, we make three other contributions. Firstly, we use a factorial ANOVA approach, which allows us to separate significant factors in a way that allows for pinpointing the most important intervention points. Secondly, the pairwise comparison allows for indicating the differences in the effects of the major intervention points. Finally, we include race as a unique characteristic of South Africa to capture the possible effects of the Broad-Based Black Economic Empowerment (B-BBEE) policy. The B-BBEE policy prioritizes black people in various areas such as procurement and ownership in a bid to redress the effects of apartheid. We surmise that because of its effect on access to various resources, this preference could affect efficiency. In addition, the inclusion of race reflects the importance of this factor on access to capital and SME performance as documented in the literature (Simatele and Kabange 2022; Robb and Morelix 2016; Bates and Robb 2016).

## 2. Materials and Methods

In developing the conceptual framework, the study adopts that SMME and lending institution characteristics affect the technical efficiency of SMMEs. This assertion is driven by the assumption that the technical efficiency of SMMEs is significantly affected by external funding success. The funding success is based on the lending institution type and the type of lending technology adopted. Furthermore, firm characteristics also influence the decision to lend. In the South African case, for example, the relationship between lending technologies, lending institutions and firm characteristics have been linked (Berger and Black 2011; Mbedzi and Simatele 2020). The interaction between these factors is also expected to amplify the effect of each variable on the level of a firm's technical efficiency.

For instance, the use of relationship lending by a development finance institution and a commercial bank is expected to have different effects on technical efficiency depending on the size of the firm because of the nature of support services that will be provided by the different types of lending institutions but also the size of the borrowing firm (Beck and Cull 2014). The firm technical efficiency phenomenon can therefore be tested empirically by determining both the independent and interaction effects of these factors (Figure 1).

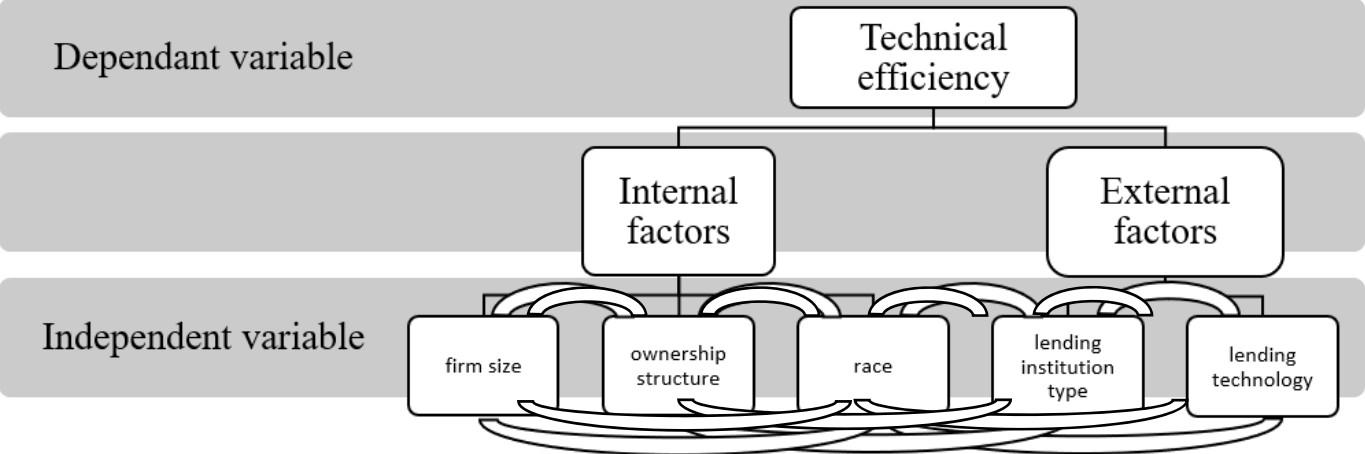

**Figure 1.** Conceptual framework for SMME efficiency.

*2.1. Data*

The study used cross-sectional survey data from 321 SMMEs. The data were collected between 28th June and 30th October 2017 in the two metropolitan municipalities; Buffalo City and Nelson Mandela Bay. The firm population comprises SMMEs in all sectors in the databases of companies compiled by the Nelson Mandela Bay Business Chamber (NMBBC) and Border-Kei Chamber of Business (BKCOB) for a list of firms operating in Nelson Mandela Bay and Buffalo City metropolitan municipalities. As of the 20th of June 2017, NMBBC and BKCOB together had a total of 1486 firms (721 and 765, respectively) with over 75% of these firms in the SMME category (NMBBC 2017; BKCOB 2017). Therefore, using Cochran's sample size for categorical data (Bartlett et al. 2001), based on the SMME population, the minimum sample size for the study was estimated to be 305 against a sample size of 321 used. Both Chambers of Business databases had firm contact details comprising company name, telephone number, email address, website, and physical address for each firm. With these details, it was easy and possible to contact the firms for appointments and physically locate firms during data gathering.

*2.2. Estimation Models*

The study adopted a three-step approach. Firstly, efficiency scores were estimated using Data Enveloping Analysis (DEA) by capturing selected activity inputs and outputs of firms using the output-oriented measure. SMEs in the African context have very limited capacity to increase inputs even when demand changes. Principally, they seek various ways of increasing output with limited inputs at their disposal. As a result, an output-oriented approach is used in the study. In the second step, a factorial analysis of variance was conducted to determine where differences significantly exist in the selected internal and external determinants of technical efficiency and their interactions, while post hoc test pairwise comparison estimations determine how much of the differences in technical efficiency exist among compared factors.

The efficiency scores were obtained by maximising the efficiency of the target SMME firm subject to the efficiency of all other firms. The scores range between 0 (least efficient)

and 1 (most efficient) (Sherman and Zhu 2006). The DEA mathematical representation is shown in Equation (1).

$$Max \ TE_j = \frac{\sum_r u_r y_{rj}}{\sum_i v_i x_{rj}}$$

$$Subject \ to : \frac{\sum_r u_r y_{rj}}{\sum_i v_i x_{rj}} \leq 1 \ for \ each \ firm, \tag{1}$$

$$and \ u_r, v_i \geq \varepsilon$$

where:

$TE_j$ = technical efficiency level of the firm $j$.

$u_r$ = weight of output $r$.

$y_{rj}$ = amount of output $r$ (recorded as annual sales in 2016 and total capital investment in year 2016 of firm $j$.

$v_i$ = weight of input $i$.

$x_{ij}$ = amount of input $i$ (recorded as number of employees by end of 2016, value of loans advanced in 2016 for firm $i$ and total value of firm assets as at the end of 2016.

Three inputs positively linked with firm output are the value of its assets, the number of employees, and the ability to attract external funds or loan advances. Literature on technical efficiency asserts the higher the value of assets of a firm the higher its technical efficiency (Amornkitvikai et al. 2014; Charoenrat and Harvie 2017). In addition, the greater the number of employees a firm has, the higher its technical efficiency (Castillo et al. 2012; Gwebu and Matthews 2018), resulting in better access to credit. In addition, firms that generate more sales are more likely to be profitable, and yet profitable firms can either reinvest their profits or access loans to create capital investments. As a result, the higher the output in terms of annual sales and capital accumulation, the higher the technical efficiency of the firm because of the link between profitability and capital accumulation to technological improvements (Martin 2001). The two important outputs positively linked to firm inputs are the value of generated annual sales and capital accumulation achieved in a trading period. Literature shows firms that generate more sales are more likely to be profitable, and yet profitable firms can either reinvest their profits or access loans to create capital investments. The above inputs and outputs determining efficiency scores are the popular firm efficiency indicators in the finance literature (Ardishvili et al. 1998; Chimucheka 2013; Delmar 1997; Mazanai and Fatoki 2012). Efficiency scores obtained from the first stage of estimation are used in the factorial analysis stage as the dependent variable. Both firm and lending institution characteristics are the explanatory variables. The aim is to determine whether the different lending institutions and firm characteristics result in differences in firm efficiency levels. Interaction effects were also computed to test whether the different firm or lending institution characteristics have a shared or unique effect on firm technical efficiency. Whenever the factorial ANOVA results are significant, differences in technical efficiency among firms exist, and the effect size is signified by Partial Eta Square. The Student–Newman–Keuls (SNK) post hoc test pairwise comparisons were then estimated on all groups with significant factorial ANOVA to determine how many technical efficiency differences exist among firm factors. The SNK was preferred because it pools the groups that do not differ significantly from each other thereby improving the reliability of the post hoc comparison by increasing the sample size used in the comparison. The factorial ANOVA mathematical representation is shown in Equation (2).

$$Y_{ij} = \mu + L_i + F_j + \gamma_{ij} + \varepsilon \tag{2}$$

where:

$Y_{ij}$ = is the technical efficiency score of each SMME.

$\mu$ = is the overall mean response.

$L_i$ = is the effect on technical efficiency due to the $i^{th}$ lending institution characteristic group level.

$F_j$ = is the effect on technical efficiency due to the $j^{th}$ firm characteristics group level.

$\gamma_{ij}$ = is the effect due to the interaction of firm characteristic and lending institution characteristics.

### 2.3. Variable Definitions

*Firm efficiency* is used as the dependent variable measured by the technical efficiency score of the firm derived using data enveloping analysis. It is a continuous variable ranging between 0 and 1. Four *types of lending technologies* are identified, which were derived from a list of concepts informing all potential lending technologies. SMMEs were asked to indicate whether these concepts applied or not in their interaction with lending institutions during the lending process. The type of technology attributed to a firm was determined by looking at the dominant lending concepts applied. For instance, if an SMME received a loan based on cash flow, profitability, or assets value information it provided the lending institution, that is, items ordinarily captured in a financial statement, then financial statement lending was used in funding that SMME. However, if the same SMME alluded to the fact that, in addition to the above requirements, it further lodged any other form of an asset as security, then asset-based lending technology was assumed, even if some of the financial statement concepts still apply. Similarly, if the SMME also asserted that, in addition to any of the above concepts, the lending institution retained part ownership of the business as part of the lending deal, then venture capital lending overrode all previous methods. Finally, if all or part of the above applied, but lending was for a specific serialized asset, then asset financing lending technology was assumed.

The *type of ownership structure* included four categories. These were sole trader–male-owned, sole trader–female-owned, family-owned, and partnership-owned businesses. A small family business is an enterprise in which the majority of the votes are held by the person who established or acquired the business (or by his or her spouse, parents, children or children's direct heirs) and at least one family member has a management or administration role in the business (Visser and Chiloane-Tsoka 2014). This definition was adopted to separate between family-owned and pure partnership-owned businesses. *Firm size* was defined as per the Small Business Act classification of small businesses in South Africa (National Small Business Act (29) of 2004). The primary items used to determine size were the number of employees, annual sales and value of assets. These are shown in Table 1. Each SMME was then fitted into the respective firm size category, as either micro, small, very small or medium. The last independent variable was the *race of the owners* of the SMME. Following the South African census race categories, four groups were identified: Black, White, Indian and Coloured[1].

**Table 1.** SMME definition by size.

| Category | Number of Employees | Annual Turnover | Total Asset Value |
| --- | --- | --- | --- |
| Medium | <200 | R31–64 m | R4.5–10 m |
| Small | <50 | R5–30 m | R1.8–4.5 m |
| Very Small | <10 | R0.15–4 m | R0.15–1.8 m |
| Micro | <5 | <R0.15 m | <R0.15 m |

Source: National Small Business Act (29) of 2004 of South Africa.

## 3. Results

The results are presented in three sets. The first is the descriptive statistics, the second is testing the significance levels for different group categories of the lending institution and firm characteristics, and lastly the final results on firm technical efficiency as a result of these factors.

### 3.1. Summary of Descriptive Statistics

The sample shows that the firms were of different ages, sizes and technical efficiency levels (Tables 1 and 2). The mean technical efficiency of firms is 61% but efficiency varies widely among firms as signified by the wide range of values of technical efficiency. Similarly, the standard deviation is very high for most factors, such as the age of the firm, the size of the firm in terms of the number of employees, the total sales volume, the value of assets, and external factors such as the value of loan advances and the length of relationships with banks. The descriptive statistics suggest that both internal and external factors could account for a wide variation in the technical efficiency levels of firms.

**Table 2.** Summary of statistics.

|  | Minimum | Maximum | Mean | Std. Deviation |
|---|---|---|---|---|
| Technical efficiency score | 0.02 | 1.0000 | 0.61 | 0.35 |
| Age of firm in years | 1 | 57 | 9.77 | 8.08 |
| Experience of owner | 2 | 53 | 14.87 | 10.12 |
| Number of employees | 2 | 192 | 24.16 | 36.44 |
| Total annual sales | 76,820 | 56,017,740 | 2,858,354.47 | 6,896,560.92 |
| Total value of assets | 5300 | 49,041,860 | 1,436,063.11 | 4,752,474.40 |
| Amount of loans advanced | 10,000 | 5,650,000 | 308,830.69 | 704,653.20 |
| Annual capital investment | 2700 | 6,491,350 | 222,196.03 | 652,340.55 |
| Total capital investment | 1000 | 9,000,000 | 361,087.50 | 987,319.30 |
| Bank length of relationship | 1 | 53 | 8.01 | 7.17 |
| Number of bank staff visits | 0 | 5 | 1.75 | 1.35 |

The technical efficiency level achieved by firms was assessed in terms of the different internal and external factors. The technical efficiency of SMMEs was measured based on the firm's activity financial inputs and outputs using Data Enveloping Analysis and ranges from a minimum score of 0 for the least to 1 for the most efficient firm.

While on average the technical efficiency of all SMMEs is 61% with a minimum of 0.19 and a maximum of 0.96, technical efficiency varies in accordance with internal and external factors (Figure 2). In terms of the sector of businesses, the least efficient firms are those in construction and engineering, while the motor industry is doing very well. However, the highest technical efficiency levels were reported from businesses that were not classified. The most technically efficient firms were financed using asset-based lending followed by asset-financing lending technology (Table 3). Venture capital lending was associated with moderate growth, while financial statement lending technology resulted in the lowest growth for SMMEs. Commercial banks grow firms better, followed by microfinance institutions, with private-owned development financial institutions lying next to government-owned development financial institutions which had the lowest technical efficiency among firms. On the other hand, technical efficiency generally declines as the size of the firm increases.

Sole trader–male- and sole trader–female-owned businesses exhibit less technical efficiency compared to partnership- and family-owned businesses. Lastly, technical efficiency also varies with race. Technical efficiency is highest among Black-owned businesses, followed by Indian, then White, while the least efficient firms are the Coloured-owned businesses. The descriptive statistics indicate that the likelihood of technical efficiency of firms is affected by several factors. These statistics support the view that there are real variations in the technical efficiency of SMMEs owing to firm characteristics (firm size, owner type, owner ethnic group, firm sector) and lending institution characteristics (lending institution type and type of lending technologies). The sections that follow assess the extent of these variations quantitatively using factorial ANOVA.

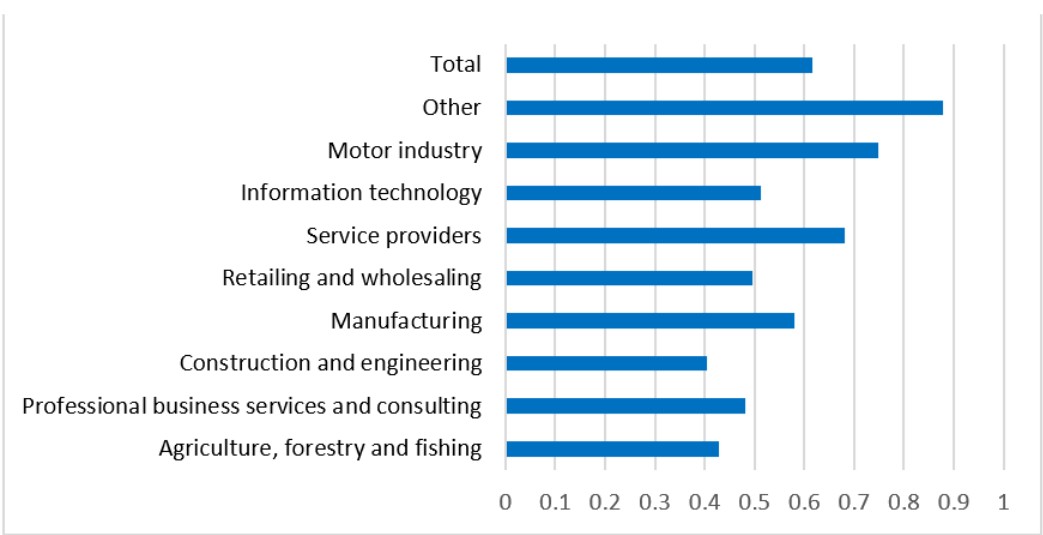

**Figure 2.** Mean technical efficiency.

**Table 3.** Technical efficiency of SMMEs.

| Determinant of Technical Efficiency | Determinant Subgroup | Mean Efficiency Score | Std. Deviation |
|---|---|---|---|
| Type of lending technology | Financial statement lending technology | 0.54 | 0.33 |
| | Asset Based lending technology | 0.71 | 0.33 |
| | Venture capital lending technology | 0.55 | 0.33 |
| | Asset financing lending technology | 0.63 | 0.37 |
| Type of lending institution | Commercial bank | 0.63 | 0.35 |
| | Government development financial institution | 0.49 | 0.29 |
| | Private development financial institution | 0.57 | 0.40 |
| | Microfinance institution | 0.62 | 0.38 |
| Size of SMME firm | Micro firms | 0.67 | 0.32 |
| | Very small firms | 0.69 | 0.34 |
| | Small firms | 0.55 | 0.35 |
| | Medium firms | 0.48 | 0.37 |
| SMME Owner Type | Sole trader–male owned | 0.60 | 0.36 |
| | Sole trader–female owned | 0.62 | 0.33 |
| | Family owned | 0.66 | 0.34 |
| | Partnership owned | 0.56 | 0.38 |
| Ethnicity of SMME Owner | Black | 0.67 | 0.33 |
| | White | 0.58 | 0.37 |
| | Indian | 0.59 | 0.31 |
| | Coloureds | 0.37 | 0.33 |
| Overall | Total | 0.61 | 0.35 |

*3.2. Significant Determinants Affecting Technical Efficiency*

The second stage uses Factorial ANOVA to indicate independent variables with significant differences in technical efficiency for both main and interaction effects. Factorial ANOVA only determines which variables have significant differences in technical efficiency. The results are shown in Table 4.

**Table 4.** Significant regressors of firm technical efficiency.

| Variable | Mean Square | DF | F-Value | Partial Eta Squared (Effect Size) |
|---|---|---|---|---|
| Model | 0.223 *** | 123 | 3.686 | 0.697 |
| Lending institution type | 0.021 | 3 | 0.345 | 0.005 |
| Lending Technology | 0.148 ** | 4 | 2.453 | 0.047 |
| Size of SMME | 0.165 ** | 3 | 2.733 | 0.040 |
| Owners' race | 0.143 * | 4 | 2.366 | 0.046 |
| Ownership type | 0.198 ** | 3 | 3.281 | 0.048 |
| Lending institution type x Lending Technology | 0.128 | 1 | 2.118 | 0.011 |
| Lending institution type x Size of SMME | 0.037 | 4 | 0.614 | 0.012 |
| Lending institution type x Owners' race | 0.256 ** | 1 | 4.241 | 0.021 |
| Lending institution type x Ownership type | 0.031 | 1 | 0.515 | 0.003 |
| Lending Technology x Size of SMME | 0.371 *** | 7 | 6.138 | 0.179 |
| Lending Technology x* Owners' race | 0.032 | 4 | 0.526 | 0.011 |
| Lending Technology x Ownership type | 0.234 *** | 9 | 3.874 | 0.150 |
| Size of SMME x Owners' race | 0.104 | 4 | 1.724 | 0.034 |
| Size of SMME x Ownership type | 0.121 ** | 8 | 2.005 | 0.075 |
| Owners' race x Ownership type | 0.113 | 5 | 1.863 | 0.045 |

Significant at [10%], (5%) and 1% = [*], (**), ***.

The factorial ANOVA model as a whole is significant, with an effect size of 70%. The type of lending technology used, the size of the SMME, the owners' race, and the type of ownership structure on their own all affect the efficiency of firms, while the type of lending institution funding an SMME on its own does not ($F$ (3, 0.021) = 0.345, $p$ = 0.793, $\eta^2$ = 0.005).

### 3.3. Effects of Lender and Firm Characteristics on the Technical Efficiency of SMMEs

In the third stage, post hoc pairwise comparison tests are used to determine the extent of the differences in the effect of the significant independent variables and their effects as estimated with the Factorial ANOVA in stage 2. After identifying which factors result in differences in technical efficiency based on significant Factorial ANOVA results, the SNK post hoc pairwise comparison test was used to explain technical efficiency differences. The post hoc pairwise comparison test gives the mean difference between groups of a factor. For example, using the firm size factor in Table 5, the mean difference (technical efficiency of firm size (I) minus technical efficiency of firm size (J)) between the technical efficiency of very-small-sized firms and medium-sized firms is 0.217 (last row). Given that technical efficiency ranges from 0 to 1, this means that, on average, SMMEs in the medium-sized class are 21.7% less efficient than those in the very-small-sized firm class.

Two separate models are presented. The first presents the main effects of each of the significant factors in Table 5. The second shows the results with the interaction effects in Tables 6–8. The results in Table 5 display the efficiency mean differences (MD) among the SMMEs and the related standard errors (SE). The results indicate that asset-based lending outperforms all other technologies, including no access to loans. SMMEs that receive lending using asset-based technologies have the highest levels of efficiency relative to other technologies. Self-financing or no borrowing has the lowest effect on efficiency relative to other lending technologies, suggesting that some form of external finance is better for efficiency than no access to external funding at all.

Firm size had a negative effect on firm efficiency. Micro-sized firms average higher efficiency levels than all bigger firms, with at least 11% higher effects on efficiency. Similarly, very-small-sized firms outperform larger firms by at least 10%. With respect to ownership type, significant differences only exist between family-owned and partnership-owned businesses. Family-owned businesses are more efficient than partnership-owned businesses.

Race, however, also affects the technical efficiency of firms. Black-owned businesses are technically more efficient than both White- and Coloured-owned businesses. The difference between Black- and White-owned businesses is the smallest, with just under 10%. The difference between Black-owned and coloured-owned businesses is higher, at 26%. Coloured-owned businesses are the least efficient.

**Table 5.** Effects of lender and firm characteristics on firm efficiency.

| 1. Lending technologies' main effects | | | |
|---|---|---|---|
| lending technology (I) | lending technology (J) | MD (I–J) | SE |
| Financial statement lending | Asset-based lending | −0.155 *** | 0.04 |
| Financial statement lending | Asset finance lending | −0.136 ** | 0.054 |
| Asset-based lending | Venture capital lending | 0.123 *** | 0.055 |
| Asset-based lending | No lending | 0.209 *** | 0.053 |
| Asset finance lending | No lending | 0.190 *** | 0.064 |
| **2. Firm size's main effects** | | | |
| firm size (I) | firm size (J) | MD (I–J) | SE |
| Micro firms | Very small firms | 0.117 ** | 0.05 |
| Micro firms | Small firms | 0.125 *** | 0.042 |
| Micro firms | Medium firms | 0.243 *** | 0.051 |
| Very small firms | Small firms | 0.100 *** | 0.044 |
| Very small firms | Medium firms | 0.217 *** | 0.053 |
| **3. SMME ownership type's main effects** | | | |
| MME Owner (I) | SMME Owner (J) | MD (I–J) | SE |
| Family-owned firms | Partnership-owned firms | 0.128 ** | 0.055 |
| **4. Owners' race's main effects** | | | |
| firm size (I) | firm size (J) | MD (I-J) | SE |
| Black | White | 0.096 *** | 0.036 |
| Black | Coloured | 0.261 *** | 0.062 |
| White | Coloured | 0.165 *** | 0.062 |
| Indian | Coloured | 0.209 ** | 0.084 |

Significant at (5%) and 1% = (**), ***.

**Table 6.** Interaction effects of lender and firm factors on efficiency—race.

| **Owners' Race and Type of Lending Institution Interaction Effects** | | | | |
|---|---|---|---|---|
| | lending Institution (I) | lending Institution (J) | MD (I–J) | SE |
| Black | Commercial bank | Microfinance institution | 0.241 ** | 0.119 |
| | Government-owned DFI | Microfinance institution | 0.246 * | 0.135 |
| White | Commercial bank | Government-owned DFI | 0.154 * | 0.084 |
| | Commercial bank | Microfinance institution | 0.154 * | 0.084 |
| | Government-owned DFI | Microfinance institution | −0.287 ** | 0.133 |
| Indian | Commercial bank | Government-owned DFI | −0.328 ** | 0.188 |
| | Commercial bank | Microfinance institution | −0.482 *** | 0.158 |
| Coloured | Commercial bank | Government-owned DFI | 0.350 *** | 0.123 |
| | Government-owned DFI | Private-owned DFI | −0.725 *** | 0.266 |

Significant at [10%], (5%) and 1% = [*], (**), ***.

These factors were interacted with each other. The results are shown in Tables 6–8. The historical reality in South Africa dictates that different racial groups have different lengths of relationships with banks. The results of the interaction analysis show that this relationship matters for efficiency. The type of lending institution on its own does not affect the technical efficiency of firms. However, once interacted with race, this effect becomes

significant. The comparison between commercial banks and microfinance institutions is significant for all races except the coloured group. For Black- and White-owned businesses, commercial bank loans have a greater effect on efficiency than microfinance loans. The effect is reversed for Indian-owned businesses. Comparing government-owned development financial institution loans and microfinance institutions shows a mixed picture. Government DFIs have a higher effect on Black-owned businesses, while the reverse is observed for White-owned businesses.

**Table 7.** Interaction effects of lender and firm factors on efficiency—firm size and ownership type.

| | 1.  Firm size and Type of lending technology interaction effects | | | |
|---|---|---|---|---|
| | lending technology (I) | lending technology (J) | MD (I–J) | SE |
| Micro firms | Financial statement lending | Asset-based lending | 0.232 *** | 0.077 |
| | Financial statement lending | Asset finance lending | −0.725 *** | 0.266 |
| Very small firms | Financial statement lending | No lending | 0.348 *** | 0.125 |
| | Asset-based lending | Venture capital lending | −0.240 ** | 0.132 |
| | Asset-based lending | No lending | 0.274 ** | 0.121 |
| | Venture capital lending | No lending | 0.514 *** | 0.166 |
| | Asset finance lending | No lending | 0.424 *** | 0.147 |
| Small firms | Financial statement lending | Asset-based lending | −0.197 *** | 0.073 |
| | Asset-based lending | No lending | 0.235 ** | 0.116 |
| Medium firms | Financial statement lending | Asset-based lending | −0.331 *** | 0.107 |
| | Financial statement lending | No lending | 0.292 *** | 0.146 |
| | Asset-based lending | Venture capital lending | 0.444 *** | 0.115 |
| | Asset-based lending | Asset finance lending | 0.518 *** | 0.134 |
| | Asset finance lending | No lending | 0.623 *** | 0.143 |
| | 2.  Firm ownership type and Type of lending technology interaction effects | | | |
| | lending technology (I) | lending technology (J) | MD (I–J) | SE |
| Sole trader–male owned | Financial statement lending | Asset-based lending | −0.212 *** | 0.069 |
| | Financial statement lending | Venture capital lending | −0.185 ** | 0.087 |
| | Asset-based lending | No lending | 0.316 *** | 0.093 |
| | Venture capital lending | No lending | 0.288 *** | 0.107 |
| | Asset finance lending | No lending | 0.207 ** | 0.1 |
| Sole trader–female-owned | Financial statement lending | Asset financial lending | −0.202 * | 0.114 |
| Family-owned firms | Financial statement lending | Asset-based lending | −0.222 ** | 0.095 |
| | Asset-based lending | Venture capital lending | 0.431 *** | 0.135 |
| | Asset-based lending | Asset finance lending | 0.323 ** | 0.127 |
| | Asset-based lending | No lending | 0.322 * | 0.132 |
| Partnership owned firms | Financial statement lending | Asset finance lending | −0.499 *** | 0.184 |
| | Asset-based lending | Venture capital lending | 0.311 ** | 0.14 |
| | Asset-based lending | Asset finance lending | −0.367 ** | 0.185 |
| | Asset-based lending | No lending | 0.320 ** | 0.138 |
| | Venture capital lending | Asset finance lending | −0.678 *** | 0.214 |
| | Venture capital lending | No lending | 0.687 *** | 0.213 |

Significant at [10%], (5%) and 1% = [*], (**), ***.

Lenders tend to vary the lending technology used depending on the size of the firm. Table 7 shows how the effect of lending technologies differs based on firm size. Asset-based and asset finance lending technologies benefit SMMEs the most across all firm sizes except for venture capital in the case of very small firms and financial statement lending for micro firms. It can be observed from the results that external financing is good for efficiency. Firms with no external financing have the lowest efficiency, regardless of firm size. Interacting

firm ownership structure and type of lending technology confirms the results which we found for firm size. Asset-based and asset-financing lending methods have the highest positive effect on efficiency, regardless of ownership type. Similarly, venture capital has greater benefits than financial statement lending and no external funding.

Finally, ownership structure was interacted with firm size, and the results are presented in Table 8. Smaller firms are more financially efficient than larger firms, except for family-owned businesses. For family-owned businesses, efficiency increases with size.

**Table 8.** Interaction effects of Ownership structure and Firm size.

| SMME Owner Type and Firm Size Interaction Effects | | | | |
|---|---|---|---|---|
| | firm size (I) | firm size (J) | MD (I–J) | SE |
| | Micro firms | Small firms | 0.139 ** | 0.068 |
| Sole trader–male owned | Micro firms | Medium firms | 0.384 *** | 0.076 |
| | Very small firms | Medium firms | 0.286 *** | 0.083 |
| | Small firms | Medium firms | 0.246 *** | 0.073 |
| | Micro firms | Medium firms | 0.444 *** | 0.121 |
| Sole trader–female-owned | Very small firms | Medium firms | 0.442 *** | 0.127 |
| | Small firms | Medium firms | 0.476 *** | 0.128 |
| | Micro firms | Very small firms | −0.297 *** | 0.102 |
| Family owned | Micro firms | Medium firms | −0.296 *** | 0.152 |
| | Very small firms | Small firms | 0.303 *** | 0.092 |

Significant at (5%) and 1% = (**), ***.

## 4. Discussion and Contributions

Our results show four main findings. First, in line with the literature, the technical efficiency of SMMEs is low and varies across sectors. The motor industry is the only one of the specific sectors that shows technical efficiencies above 80%. These low levels of technical efficiency can be explained by low capital-labour ratios, low scale (Padmavathi 2019; Xaba et al. 2018), poor skills (Padmavathi 2019; Charoenrat and Harvie 2013) and technology (Fahmy-Abdullah et al. 2021). These characteristics are very typical of SMEs in Africa.

Secondly, technical efficiency is negatively correlated with firm size. Our results show that while most SMMEs are generally inefficient, micro-firms are the most efficient, meaning efficiency is inversely linked to firm size. Similar results have been reported by Aggrey et al. (2012) and Radam et al. (2008). While firm size matters, its effect on technical efficiency is influenced by the ownership structure of the firm. Smaller SMMEs are more technically efficient than larger ones, but that effect is increased by the shift from individually owned to multiply owned firms. However, when individually owned firms are compared among themselves, sole trader–female-owned firms are more technically efficient than sole trader–male-owned businesses, implying a gender effect. Although this result was observed in the literature, the typical result is a positive relationship between size and technical efficiency.

Our study disaggregates the effect of size by ownership type. This disaggregation and the negative result could be evidence of the firm's ability to successfully manage its input constraints (see Agostino et al. 2018). The results suggest that the direct control of results by the owner can lead to increased efficiencies. This might seem counterintuitive at first. However, when the size of the firm and resource constraints available to microenterprises for the day-to-day output is considered, it is clear that they can run their businesses on very few resources minimizing the opportunity for waste.

Thirdly, our results reveal that access to external funding has a positive effect on efficiency. In the literature (Barchue and Aikaeli 2018), the technical efficiency of firms was found to be associated with access to finance. Furthermore, financial assistance from the government had a high contribution to the technical efficiency of SMMEs (Charoen-



rat and Harvie 2017). Whereas access to finance improves technical and technological expert intensity (Le and Harvie 2016), the availability of funded machinery and equipment (Sekonopo et al. 2017) also contributes to the technical efficiency of firms. The literature suggests that differences in industry characteristics affecting financing could explain these results. Our study focuses on the effects of finance especially as it relates to sources of funding and the technology used to provide that funding. This highlights our next finding, that external financing has a positive effect on efficiency. We find that the type of lending technology affects efficiency, but that effect is influenced by both ownership structure and firm size. For lending technologies, secured lending exhibits high technical efficiency than unsecured lending methods for all firms but the level of technical efficiency is enlarged as firm size increases. Additionally, the technical efficiency is magnified as ownership shifts from individually owned firms to firms with multiple owners. On average, firms which use external sources of funding are more technically efficient than those funded by internal sources or no funding regardless of the type of lending technology used or its interaction with either firm size or ownership structure. Furthermore, in line with Harvey, we find that the benefits of the impact of size on efficiency are influenced by the source of funding. More targeted funding technologies such as asset financing and venture capital offered by development financial institutions have a greater effect on efficiency among smaller firms. This suggests that targeted financing programmes can help SMMEs become more efficient and resilient relative to sourcing financing from commercial banks and microfinance institutions. Nonetheless, the effects of lending technologies on efficiency are scanty in the literature. So far, only one study (Agostino and Trivieri 2019) has identified the relationship between lending technologies and efficiency. However, the study assessed the effect of trade credit lending technology only. While trade credit improves the efficiency of SMMEs, the study does not show how the effects differ if different types of lending technologies are used. Our paper covers that gap.

Finally, our results find the effectiveness of different types of sources of capital is affected by the race of the business owner. We find that while the type of lending institution on its own does not affect the financial efficiency of firms, different types of lending institutions have different financial efficiency levels for firms owned by different races. Therefore, funding to firms by different types of lending institutions is highly associated with the race of the owners of the firms, resulting in different technical efficiency levels. High levels of technical efficiency for Black-owned firms are linked to commercial banks and government-owned development financial institutions. In contrast, for White-owned firms, technical efficiency is high with commercial banks, for Indian-owned firms, it is linked with microfinance institutions and government-owned development financial institutions, while for Coloured-owned firms, it is linked with private-owned development financial institutions and commercial banks. Race, therefore, dictates the type of lending institutions more likely to fund the owner's firm, and the resultant technical efficiency that follows as a product of that funding. Gender, therefore, affects the technical efficiency of SMMEs. These findings support the role of the Broad-Based Black Economic Empowerment (B-BBEE) program, which focuses on improving income equality throughout the whole economy. A scan of previous studies shows the effects of both lending technologies and lending institutions are missing in the extant literature, which is a major contribution of this paper.

As a result, the paper identifies three major contributions to extant literature. Firstly, the paper adopts a factorial ANOVA approach that separates significant factors in a way that allows for pinpointing the leading intervention points. Secondly, the pairwise comparison allows for an indication of the differences in effects; and finally, by including race as a unique characteristic of South Africa, it captures well the possible effects of the BEE policy on efficiency given that the program prioritises income equality in the whole economy. This study used cross-sectional data. Future studies could use panel data to capture the effects of dummy variables like the Broad-Based Black Economic Empowerment policy on the technical efficiency of small businesses as an effective intervention measure to address inequality problems in South Africa.

**Author Contributions:** The full manuscript was developed by the first author as part of their postdoctoral research fellowship work under the mentorship of the second author. Therefore, both authors contributed equally to this work. All authors have read and agreed to the published version of the manuscript.

**Funding:** This research was funded by Research Board of the National University of Science and Technology under grant number RB/132/16 And The APC was funded by the Govan Mbeki Research and Development Centre (GMRDC) of the University of Fort Hare.

**Informed Consent Statement:** Informed consent was obtained from all subjects involved in the study. In addition, an ethical clearance certificate number SIM031SMBE01 dated 1 June 2017 was issued in respect of this study by the University of Fort Hare's Research Ethics Committee (UREC).

**Data Availability Statement:** The data presented in this study are available on request from the corresponding author. The data are not publicly available because authors still need to use it future.

**Conflicts of Interest:** The authors declare no conflict of interest.

## Note

[1] Coloured people in South Africa refers to mixed race individuals. They make up about 9% of the South African population STATSSA (2019).

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
