# Peer review of "Lending Technologies, Firm Characteristics and Small Business Efficiency in South Africa"

_economies, doi:10.3390/economies10110289_

Round 1
Reviewer 1 Report
The article and its subject matter are interesting. There are only a few issues that would need to be addressed before publication:
- An explanation of why the output-oriented model was used would have been useful. In the case of companies, they often have no influence on demand and what they can adjust is costs - i.e. input. In this case I understand that technology improvements are involved, i.e. outputs will improve, but it would still be useful to have an explanation in the text.
- Line 176 - are these inputs positively linked to efficiencies, or only to outputs?
Reviewer 2 Report
1. The concept of lending technology is neither explained in the introduction/literature nor in the method & material Section
2. How can lending technology/technologies affect the efficiency of SMMEs? No relationship is hypothesized.
3. Authors have declared that the amount of lending is not as important as the source of lending. How?
4. Authors need to Improve the readability of the last paragraph of the review of literature starting from "Literature Submits........."
5. Technique of selecting the sample size (321 SMMEs) is also not clear.
6. Title of the manuscript is very ambiguous. It reflects all the determinants of technical efficiency of SMMEs but the manuscript talks mainly about the relationship of financial institutions with efficiency. (if possible, the Authors must bring more clarity to the title of the manuscript.
Reviewer 3 Report
A major issue is the fact that the autors introduce RACE as variable. They have to pay attention to possibile complaints from the readers. In addition, there are no previous studies cited that justify the use of RACE as possible explanatory variable. Therefore WHY RACE?
The second key issue is the fact that efficiency is measured/scored using size. The authors use inputs that the literature indicates being positively linked with firm efficiency:: value of assets, no employees, and the ability to attract external funds. At least two of them measure the size of a firm. So you use size to measure the efficiency while your research question is investigating if SIZE is related to effienciecy. It seems a Recursive Reasoning.
You have to highlight this aspect as great limitation.
In addition some information on outputs detected thorugh the questionnnaire seem missing. The authors write "The above inputs and outputs determining efficiency scores" but only inputs (assets, employees, loans) are described which can be classified as inputs and not outputs
The third point is on how you classify the type of ownership structure. Could you please cite references/previous work that use this classification? According to my understanding they are not mutually exclusive. For example a family-owned business can be also a partnership-owned businesses, just the family members are partners/owing the company.
Some minor comments
- Introduction:
I would update or delete the very old reference to 2003 and 2007 ("World Bank’s 26 2003 and 2007 SMMEs surveys in South Africa showed SMMEs have low technical effi- 27 ciency due to high factor unit costs"
I would delete the relationship/discussion of how ownership structure impacts on risk taking....risk taking is not related to technical efficiency ("Ownership structure affects the efficiency of the firm. The first channel of transmis- 52 sion is risk-taking. The literature shows that risk appetite differs depending on the own- 53 ership structure. Sole proprietors are risk averse and are more likely to invest in low-risk 54 ventures than firms with multiple owner")
- I would list the three additional variables the authors mention on page 2 line 65, where they state "Our paper introduces three additional variables. The first variable ..." BUT only the first variable is mentioned and explain. Here it is necessary to ancitipate the other variables (race,....) You can use The first variable is...because....The second variable is.... because (in a very short but effective manner)
- Final section
there are no limitations nor indications for future research
Round 2
Reviewer 2 Report
Thanks for including all the suggestions!